# Implications of Minimum Description Length for Adversarial Attack in Natural Language Processing

**DOI:** 10.3390/e26050354

**Published:** 2024-04-24

**Authors:** Kshitiz Tiwari, Lu Zhang

**Affiliations:** Department of Electrical Engineering and Computer Science, University of Arkansas, Fayetteville, AR 72701, USA; ktiwari@uark.edu

**Keywords:** minimum description length, adversarial attack, NLP, causal learning

## Abstract

Investigating causality to establish novel criteria for training robust natural language processing (NLP) models is an active research area. However, current methods face various challenges such as the difficulties in identifying keyword lexicons and obtaining data from multiple labeled environments. In this paper, we study the problem of robust NLP from a complementary but different angle: we treat the behavior of an attack model as a complex causal mechanism and quantify its algorithmic information using the minimum description length (MDL) framework. Specifically, we use masked language modeling (MLM) to measure the “amount of effort” needed to transform from the original text to the altered text. Based on that, we develop techniques for judging whether a specified set of tokens has been altered by the attack, even in the absence of the original text data.

## 1. Introduction

Recent research has shown that language models are generally not robust and do not generalize well when exposed to out-of-distribution data. This makes them vulnerable to various types of adversarial attacks including spurious tokens, typos, and word order changes [1,2,3]. These vulnerabilities are significant concerns as language models are increasingly used in various applications. To address these challenges, researchers are exploring methods to develop more robust and reliable language models by improving the diversity and representativeness of training data as well as using techniques such as adversarial training and regularization to enhance model robustness [4].

In the domain of adversarial attacks on machine learning models, a key area of interest lies in the exploration of causality to develop new criteria for learning robust natural language processing (NLP) models that go beyond exploiting correlations only. Existing endeavors in using causal reasoning to help build robust NLP models mainly focus on dealing with spurious correlations [5,6]. In these works, the words that may be targeted by attacking models are viewed as confounders that introduce spurious correlations between the features and the label. Popular approaches for mitigating spurious correlations include data augmentation [7,8,9,10] and multi-environment training [11], where the former generates counterfactual instances (e.g., replacing keywords with their synonyms) and incorporates them into the training data, and the latter attempts to train the model or representation function that is invariant across multiple environments. Although these approaches are shown to be effective in many cases, they still present certain challenges and limitations. For example, for data augmentation, it is difficult to identify the keywords and their replacement. It has been shown that new spurious correlations may be introduced if the keyword lexicons are incomplete [12]. For multi-environment training, obtaining data from multiple labeled environments could be quite challenging.

In this work, we study the problem of robust NLP from a complementary but different angle: as attack models are purposefully designed systems that exploit vulnerabilities to produce inaccurate predictions or classifications, we treat the behavior of attack models as complex causal mechanisms. Our endeavor delves into understanding how the activities of these adversarial models can be quantified in terms of causal influence. Specifically, we aim to discover the causal relationship between the original text data and the modified text data produced by the adversarial model. Building on that, we make a preliminary effort to identify the actions of these adversarial models, that is, to detect if a specific set of tokens has been altered.

Causal discovery has garnered significant attention across diverse datasets and contexts. Studies exploring causal discovery have been conducted for both static data [13] and temporal data [14,15]. However, a noticeable research gap exists in relation to NLP data. Inspired by [16] that demonstrates the validity of the principle of independent causal mechanisms (ICMs) in NLP, we propose to investigate causal relationships between tokens by utilizing the minimum description length (MDL) framework [17]. Our method is particularly centered around the use of masked language modeling (MLM) [18] for measuring the “amount of effort” needed to predict one set of tokens from another set of tokens, given the context (see Figure 1). Specifically, we postulate a causal graph as shown in Figure 2 which presents different components of a text in a causal relationship. Here, *B* represents a set of tokens altered by the attacking model, and *A* represents the original tokens of *B*, satisfying the condition: |A|=|B|, A∩B=∅ and they have shared positions. The arrow from *A* to *B* signifies the causal influence exerted by the attacking behavior. Additionally, *C* encompasses the remaining words existing within the text excluding *A* and *B*, encapsulating the contextual background of the text. We presume the arrows from *C* to *A* and *B*, which denote the contributory role of the remaining text words towards the causal dynamics underpinning both *A* and *B*. As a result, the causal graph captures the interplay between modified tokens, original tokens, and contextual cues, forming the basis for comprehending the intricate behavior induced by adversarial interventions. Based on that, MDL is computed as the code-length of compressing A,B given *C*, which reflects the algorithmic information presented in the attacking model. In addition, to handle practical situations where the original text data are absent, we propose to generate proxies of the original tokens by using LLMs for the MDL computation. The experiment results show that the MDL computed is propositional to the fraction of the modified tokens contained in *B*, which can serve as an indicator of detecting adversarial attacks.

Our work creates innovative techniques that are complementary to existing robust NLP methodologies. For example, the work by [10] showcases the feasibility of mitigating the influence exerted by adversarial tokens, thus formulating a resilient hate speech detection model. However, their study is based on prior knowledge of adversarial tokens—an assumption that diverges from the intricate dynamics of real-world scenarios. Our method has the potential to identify the adversarial tokens within the training data, enhancing the applicability of the existing robust NLP models.

## 2. Related Work

### 2.1. Adversarial Attacks in NLP

Adversarial attacks in natural language processing (NLP) are strategic perturbations to textual input aimed at deceiving machine learning models while maintaining human interpretability. Adversarial attacks can be typically categorized into character-level, word-level, and sentence-level attacks. At the character level, adversarial attacks utilize subtle character manipulations, including insertions, deletions, and swaps, often blending natural spelling mistakes with synthetic noise, like keyboard neighbor substitutions, randomization, and punctuation adjustments. Such methods, illustrated by DeepWordBug [19] and TextBugger [20], monitor the edit distance to ensure sentence readability. At the word level, adversaries exploit techniques like gradient-based [21], importance-based [22,23], and replacement-based strategies [24]. These approaches manipulate whole words, leveraging gradients to reverse classification probabilities or replacing words with semantically coherent alternatives using word vectors. Sentence-level attacks, on the other hand, alter groups of words, introducing techniques like ADDSENT and ADDANY [25,26] and utilizing GANs to generate semantically consistent sentences, as seen in AdvGen [27]. Altogether, these diverse and intricate adversarial strategies highlight the significant challenges and vulnerabilities in ensuring the robustness of NLP models against malicious manipulations [28]. In this paper, we mainly target character-level and word-level attacks and employ BERT-ATTACK [22] and TextFooler [23] as the attacking models for our analysis.

### 2.2. Robust NLP

Ensuring robustness in natural language processing (NLP) is an important task. Typical approaches encompass defenses against adversarial attacks and techniques to enhance model generalization and reduce sensitivity to input variations. Data augmentation strategies, such as generating counterfactual instances and integrating them into training data, are widely adopted to enhance model invariance and sensitivity by incorporating perturbations to confounding factors [7,29,30]. To mitigate the impact of perturbations to these confounding factors on the predictions, a penalty term can be introduced in the objective function [10]. These counterfactuals can be generated through a variety of methods: manual post-editing [7], which offers fluency but is time-consuming; keyword replacement [8,29], which may lack generalizability; and automated text rewriting [9], which aims to balance fluency with efficiency. While counterfactuals are a powerful tool, especially in addressing the gaps inherent in causal inference, their generation poses challenges. It can be difficult to create meaningful and precise counterfactuals, and there is a risk of introducing new spurious correlations [31].

As an alternative to data augmentation, robust learning algorithms can be designed directly using observed data. One strategy for invariance tests involves ensuring that the model adopts to a specific distributional criteria. For instance, counterfactually invariant predictors can be achieved by considering the causal structure of the data-generating process [32]. This approach formalizes counterfactual invariance to certain factors, ensuring that predictions are unaffected by particular attributes [16]. Observing the true causal structure of the problem can help derive observed-data signatures of counterfactual invariance, which can be incorporated as regularization terms in training. Alternatively, training data can be viewed as originating from different environments, each with distinct cause distributions but a shared causal relationship. Environmental invariance criteria require an invariant representation function to ensure uniform predictor performance across causally compatible domains, known as domain generalization [33,34]. Both approaches necessitate richer training data compared to standard supervised learning, posing challenges, especially when explicit labels or data from multiple labeled environments are required [31].

## 3. Background and Preliminaries

### 3.1. Masked Language Modeling (MLM)

Masked language modeling [18] involves the task of predicting a specific token within a given context. To formally describe this process, let us consider an input sequence denoted as *X*, comprising a series of *K* tokens represented as $\left\{x_{1},\ldots ,x_{K}\right\}$. For ease of understanding, let $x_{i}$ symbolize the *i*-th token within sequence *X*, and let $X_{\left[i\right]}^{mask}$ represent the sequence obtained by substituting the *i*-th token with a special token denoted as [*MASK*]. In a more precise manner, the expression $X_{\left[i\right]}^{mask}$ corresponds to the following sequence:$$X_{\left[i\right]}^{mask}=\left\{x_{1},\ldots ,x_{i-1},\left[\mathrm{MASK}\right],x_{i+1},\ldots ,x_{K}\right\}$$

Additionally, we define the sequence $X_{\left[i,j\right]}^{mask}$ as the result of replacing both tokens $x_{i}$ and $x_{j}$ with the [MASK] token. For a sequence with multiple tokens replaced with [MASK] token, we denote it as $X^{mask}$. The objective of MLM is to acquire a probabilistic model $p_{\theta }$, where θ denotes the model’s parameters, that learns to predict the masked tokens in $X^{mask}$ given *X*, the input sequence. The final output of the model is a sequence of tokens $X^{pred}$, where the [MASK] tokens are replaced with the most probable tokens based on the acquired probabilistic model $p_{\theta }$. This model is intended to capture the underlying patterns and relationships in the language data, thereby enabling the accurate prediction of masked tokens within various contexts. The inherent design of the masked language model ensures that the produced sentences are coherent, grammatically accurate, and maintain most of the semantic meaning [22].

The evaluation of the MLM task’s effectiveness can be accomplished through an analysis of its associated loss function: ${\mathrm{Loss}}_{i}=-\sum_{j}y_{i,j}\log\left({\hat{y}}_{i,j}\right).$ Here, ${\hat{y}}_{i}$ represents the anticipated probability distribution for the token at the *i*-th position, whereas $y_{i}$ signifies the target distribution encoded in one-hot fashion for the said token, and *j* indexes each token in the model’s vocabulary, allowing the loss calculation to assess the model’s predictions against the actual token for each position *i* where a token was masked. The cumulative loss for all instances where tokens have been replaced by the [*MASK*] token within the $X^{mask}$ sequence is computed as the sum of these individual losses, succinctly expressed as: ${\mathrm{Loss}}_{X}=\sum_{i\;\mathrm{is}\;\mathrm{masked}}{\mathrm{Loss}}_{i}.$

### 3.2. Minimum Description Length (MDL)

The minimum description length (MDL) concept operates as a statistical framework that balances the intricate interplay between model complexity and its alignment with observed data. This is achieved by merging the data and the generative model into a single entity, resulting in a unified encoding. The optimal model choice is determined by minimizing the resulting encoding length [17,35].

The fundamental principle of MDL asserts that the model producing the shortest encoding length for the data is better equipped to capture the underlying structures within the data [17]. For a given set of variable pairs ${\left\{\left(x_{i},y_{i}\right)\right\}}_{i=1}^{n}∼P_{Y,X}$, MDL provides a concise bit-length for compression along with the necessary parameters for subsequent decompression. This inherent fusion of intricacy and alignment confers upon MDL a powerful utility for both selecting models and analyzing data. In cases involving two independent variables, represented as *X* and *Y*, the computation of MDL is as follows:(1)MDL(X,Y)=MDL(X)+MDL(Y|X)=MDL(Y)+MDL(X|Y),
where the notation MDL(·|·) denotes the conditional compression in which the second argument is regarded as “free parameters” and does not contribute to the compression length of the first argument. Equation (Equation 1) can be interpreted as a comparison between two methods for compressing matching data points (*X*, *Y*). In the first approach, *X* is compressed first, followed by the compression of *Y*, which is conditional on *X*. In the second approach, *Y* is compressed first, followed by the compression of *X*, where the conditioning is rooted in *Y*.

### 3.3. Causal Relation and MDL

Causal graphs are a widely utilized framework for representing and deducing causal relationships among variables. Typically structured as directed acyclic graphs (DAGs), denoted as G={V,E}, these graphs consist of a set of variables *V*, and *E* depicts the causal connections between them [36]. A directed edge stemming from variable *X* and pointing to variable *Y* signifies that *X* acts as the cause, exerting a direct influence on *Y*. The integration of causal graphs in activities such as causal representation, inference, and reasoning is pervasive [37].

In instances where *X* is the causal factor and *Y* is the effect, the determination of the likelihood of *Y* given *X* corresponds to the causal direction (X→Y). Conversely, evaluating the probability of *X* given *Y* represents the anticausal direction (Y→X). The authors in  [38] introduced the concepts of “causal learning” (X→Y) and “anticausal learning” (Y→X) to describe learning in the respective directions. For a given pair of variables *X* and *Y*, if the generative processes $P_{X}$, $P_{Y}$, $P_{X|Y}$, and $P_{Y|X}$ are mutually independent—adhering to the independent causal mechanism (ICM) principle [38]—the minimum description length (MDL) framework can be employed to compute causal direction (X→Y), which is inherently less than or equal to the computation of the anticausal direction (Y→X) [16].

## 4. Methodologies

Our objective is to detect the algorithmic information of the attacking model (e.g., BERT-ATTACK) as causal influences given the modified text data. To this end, we first consider any two sets of tokens A,B and a context *C* and devise the technique of quantifying the ‘amount of effort’ needed for predicting *B* from *A* given *C* by using MDL. We view this quantification as the algorithmic information of the mechanism transitioning from *A* to *B*, i.e., (A→B|C) (e.g., when *A* and *B* are original and modified tokens, this arrow represents the mechanism of the attacking model). Then, based on that, we develop techniques for judging whether a specified set of tokens has been modified by the attacking model, even in the absence of the original text data.

### 4.1. Attacking Model

In this paper, we employ BERT-ATTACK [22] and TextFooler [23] as the attacking model. This methods are used for conducting adversarial attacks on text models.

#### 4.1.1. BERT-ATTACK

BERT-ATTACK leverages the BERT language model to create adversarial texts for black-box untargeted attacks in machine learning. In these attacks, the attacker does not have information about the target model’s structure, parameters, or gradients. The technique involves identifying key tokens that significantly impact the model’s prediction. These tokens are then replaced with semantically similar and grammatically correct alternatives until the model’s prediction is altered. The approach exploits the characteristics of the original BERT model to deceive a fine-tuned BERT model, encompassing two primary steps: identifying influential tokens and replacing them to change the prediction.

Identification of Vulnerable Words:

In the first step, the method assigns an importance score, denoted as $I_{w_{i}}$, to each word or token in the text. This score is determined by comparing the logit output of the target model when using the original text and when using the same text with the word $w_{i}$ masked. The importance score $I_{w_{i}}$ is formally defined as:$$I_{w_{i}}=o_{y}\left(S\right)-o_{y}\left(S_{\setminus w_{i}}\right),$$
where $S=[w_{0},\cdots ,w_{i},\cdots ]$ denote the input sentence, and $o_{y}\left(S\right)$ represents the logit output of *S*. Similarly, $S_{\setminus w_{i}}=\left[w_{0},\cdots ,w_{i-1},\left[\mathrm{MASK}\right],w_{i+1},\cdots \right]$ denotes the sentence with the word $w_{i}$ masked, and $o_{y}\left(S_{\setminus w_{i}}\right)$ represents the logit output respectively. The words or tokens are then ranked based on their importance score and the top *x* most vulnerable words are selected.

Word Replacement:

After identifying the list of vulnerable words, the method iteratively replaces these words to discover perturbations that can confuse the target model. Notably, this process ensures that the generated sentences remain both semantically coherent and grammatically accurate by using the BERT’s masked language model. To enhance diversity, the method considers the *K* most probable predictions at each position, as opposed to relying solely on the argmax prediction from the BERT model.

Additionally, because BERT uses Byte-Pair Encoding (BPE) for tokenization, word substitutes for tokens with sub-word components are chosen based on the perplexity of sub-word combinations, ensuring that suitable word replacements are selected at the sub-word level. This comprehensive approach results in a tactful execution of the attack, with the ultimate aim of compromising the target model’s performance without compromising the attack sentence’s semantic and grammatical structure.

#### 4.1.2. TextFooler

Similar to BERT-ATTACK, TextFooler initiates its adversarial process by identifying and modifying key tokens, aiming to alter the predictive output of the model. The process involves two main steps:Identification of Vulnerable Words:

The initial stage of this process involves using a selection technique to assess the impact of individual tokens on the model’s prediction. This is achieved by assigning an importance score, denoted as $I_{w_{i}}$, for each word $w_{i}$ in the input sequence $X=\{w_{1},w_{2},\cdots ,w_{n}\}$. The importance score quantifies the effect of word $w_{i}$ on the model’s prediction. The calculation of $I_{w_{i}}$ is based on the variation in prediction when the word $w_{i}$ is removed. This is formally expressed as:$$I_{w_{i}}=\left\{\begin{array}{c} F_{Y}\left(X\right)-F_{Y}\left(X_{\setminus w_{i}}\right),\;\mathrm{if}\;F\left(X\right)=F\left(X_{\setminus w_{i}}\right)=Y,\\ \left(F_{Y}\left(X\right)-F_{Y}\left(X_{\setminus w_{i}}\right)\right)+\left(F_{\bar{Y}}\left(X_{\setminus w_{i}}\right)-F_{\bar{Y}}\left(X\right)\right),\\ \;\mathrm{if}\;F\left(X\right)=Y,F\left(X_{\setminus w_{i}}\right)=\bar{Y},\mathrm{and}\;Y\neq \bar{Y}. \end{array}\right.$$
where, $F_{Y}\left(\cdot \right)$ denotes the model’s prediction score for label *Y* on some input, and $X_{\setminus w_{i}}=\left\{w_{1},\cdots ,w_{i-1},w_{i+1},\cdots ,w_{n}\right\}$ represents the sequence obtained after removing the word $w_{i}$ from *X*.

Word Replacement:

After identifying the important words, this process involves two primary steps: selecting suitable synonyms and ensuring semantic and grammatical alignment with the original sentence. In the first phase, Synonym Selection and POS Alignment, the strategy begins by identifying potential synonyms for vulnerable words using a pre-trained embedding model [39]. Using these embedding vectors, top N synonyms were selected based on their cosine similarity with a word *w*, exceeding a set threshold. This method aims to strike a balance between synonym diversity and maintaining semantic similarity, focusing on the most semantically similar synonyms for word replacement. The process then includes a Part-of-Speech (POS) check, retaining only synonyms that match the POS of the original word, thus preserving grammatical consistency.

In the second phase, Semantic Evaluation and Finalization, the next step involves assessing the semantic similarity between the original and modified sentences using the Universal Sentence Encoder (USE) [40]. Synonyms exceeding a set similarity threshold are shortlisted. The final selection of a synonym is guided by its potential to alter the target model’s prediction or by having the lowest confidence score for the original label. This iterative approach is continued until the prediction is modified, ensuring that the adversarial examples are both effective in deceiving the model and semantically similar to the original text.

### 4.2. MDL for Inferring Causal Direction between Tokens

Consider *X* as the tokenized input sequence. Let A,B be two disjoint subsets of tokens in *X* and let C=X∖{A,B} be the context. By assuming the causal graph in Figure 2, the MDL for compressing A,B given *C* is computed as
MDL(A,B|C)=MDL(A|C)+MDL(B|A,C)≤MDL(B|C)+MDL(A|B,C),
where, [MDL(A|C)+MDL(B|A,C)] is the MDL of encoding MDL(A,B|C) in causal direction and [MDL(B|C)+MDL(A|B,C)] is for anticuasal direction.

The independent causal mechanisms (ICM) principle is a fundamental concept in causal inference, stating that in any complex system, causal processes operate independently of one another. According to the ICM principle, the MDL computed by following the causal direction, i.e., first compressing *A* given *C* and then compressing *B* given A,C should be smaller than the MDL computed vice versa, which leads to the above inequality.

To measure the quality of causal relations in the data, we consider a normalized metric to ensure a fair comparison between different data sizes and randomness. Following the methodology presented by [41], we define the causal indicator $\Delta_{A\to B}$ as follows:$$\Delta_{A\to B}=\frac{\mathrm{MDL}(A|C)+\mathrm{MDL}(B|A,C)}{\mathrm{MDL}(A|C)+\mathrm{MDL}(B|C)}.$$

The numerator denotes the combined description length for the causal direction, reflecting the ‘amount of effort’ to predict *B*. On the contrary, the denominator acts as an equalizer, considering the individual description lengths needed to predict both *A* and *B* with the textual context *C*. Intuitively, this metric gauges the relative efficiency of describing *B* when *A* is known, compared to when it is not. Similarly, we also define the causal indicator $\Delta_{B\to A}$, which evaluates the potential causal relationship in the reverse direction. Finally, a confidence score *S* based on [41] is defined as:$$S=|\Delta_{A\to B}-\Delta_{B\to A}|,$$
which serves as a measure of the strength of the causal relationship between *A* and *B*. A higher confidence score indicates a stronger causal relationship, while a score close to 0 suggests the absence of any causal relationship between the two sets of words.

### 4.3. Calculating MDL with MLM

We compute the code-length for MDL(A,B|C) by leveraging masked language modeling (MLM) which revolves around determining the optimal token replacement for the designated mask token. Initially, the BERT’s masked language model is fine-tuned using the context *C* to enable a comprehensive assessment of code-length for both directions. Throughout this fine-tuning process, the positional integrity of *A* and *B* is maintained. This is facilitated through the introduction of a specialized token, denoted as [*CAUSAL*], which serves to preserve the positional information of *A* and *B*. This strategy is essential as our objective is to refine the model’s ability to predict *A* independently of *B*. Therefore, to clearly distinguish the roles of *A* and *B*, we employ two different special tokens: [MASK] and [CAUSAL]. The introduction of the [*CAUSAL*] token plays a pivotal role in making the model aware of the positional importance of these variables.

The resultant fine-tuned model, denoted as $p_{\theta }^{f1}$ subsequently serves as the foundation for calculating the code-length of the conditional terms and conditional joint terms. The adoption of the fine-tuned model offers the advantage of avoiding the necessity to compute the marginal term MDL(C) through alternative language models while simultaneously ensuring the effective transmission of contextual information *C* in the computation.

Next, we extend the methodology outlined by [42] from its original application in classification to the context of MLM. As a result, the computation of the MDL can be given by Equation (Equation 2):(2)$${\mathrm{MDL}}^{\mathrm{online}}\left(A|C\right)=t_{1}\log_{2}K-\sum_{i=1}^{n-1}\log_{2}p_{\theta_{i}}\left(A_{t_{i}+1:t_{i+1}}^{out}|C_{t_{i}+1:t_{i+1}}^{in}\right),$$
where $t_{1}$ is the length of the first subset of data, and *K* is the fixed number of tokens, set to 8000 for computational uniformity, for which a uniform code length is maintained. This setup ensures that the initial segment of the data is encoded with a uniform probability distribution over *K* possible tokens, reflecting a non-informative prior or a starting point in the absence of specific data-driven probabilities. The term $p_{\theta_{i}}$ represents the probability distribution parameterized by $\theta_{i}$, which models the likelihood of observing a certain sequence of output tokens $A_{t_{i}+1:t_{i+1}}^{\mathrm{out}\;}$ given the contextual input $C_{t_{i}+1:t_{i+1}}^{in}$ for each interval from $t_{i}+1$ to $t_{i+1}$. Here, $\theta_{i}$ denotes the set of parameters that define the model’s state at the *i*-th interval, encapsulating the learned relationships between input contexts and output token sequences up to that point in the sequence.

To compute MDL(A|C) (and similarly for MDL(B|C)), the input sequence denoted as $C^{in}$ is the token sequence with *A* masked by [*MASK*] tokens and *B* replaced by [*CAUSAL*] tokens. The corresponding out, denoted as $A^{out}$, is the sequence obtained from $C^{in}$ by substituting the [*MASK*] tokens with their predicted counterparts in *A*. Then, following the online coding method [42], we partition the dataset into *n* non-overlapping subsets, each increasing in size (in our experiments, we use n=10 and the size percentages of the 10 subsets are 0.1, 0.2, 0.4, 0.8, 1.6, 3.2, 6.2, 12.5, 25, and 50, respectively). These subsets are defined by their ending indices, represented as $t_{1}$ through $t_{n}$. Similarly, we can obtain the token subsets of $C^{in}$ and $A^{out}$, where the *i*-th subsets are denoted as $C_{t_{i-1}+1:t_{i+1}}^{in}$ and $A_{t_{i-1}+1:t_{i+1}}^{out}$. We then iteratively fine-tune and evaluate the model on the data. For the initial time step, we employ the previously obtained fine-tuned model $p_{\theta }^{f1}$ to ensure that we have a model equipped with contextual information from the text. We fine-tune the model on the first subset of $A^{out}$ (denote the fine-tuned model as $p_{\theta_{1}}^{f1}$) and compute the code-length on the second subsets by evaluating the log-likelihood $\log_{2}p_{\theta_{1}}^{f1}\left(A_{t_{1}+1:t_{2}}^{out}|C_{t_{1}+1:t_{2}}^{in}\right)$. Then, in each following time step *i*, we fine-tune the model on the aggregated data of $A^{out}$ from subset 1 to subset *i* based on the previous model $p_{\theta_{i-1}}^{f1}$ to obtain $p_{\theta_{i}}^{f1}$, and compute the code-length on the i+1 datasets, until the last time step *n*. We denote the final fine-tuned model $p_{\theta_{n-1}}^{f1}$ as $p_{\theta }^{f2}$, which serves as the initial point for calculating the MDL of the conditional joint terms in the next step.

To compute MDL(B|A,C) (and similarly for MDL(A|B,C)), the input sequence is the token sequence with *B* masked by [*MASK*] tokens, and the output sequence is the original token sequence where the [*MASK*] tokens are substituted by the original tokens in *B*. We follow a similar data partitioning approach where the model is iteratively fine-tuned on one subset of the data and evaluated on another subset of the data, using $p_{\theta }^{f2}$ as the initial model. The ${\mathrm{MDL}}^{\mathrm{online}}\left(B,A|C\right)$ is computed similarly to Equation (Equation 2). Finally, the MDL(A,B|C) following the causal direction shown in Figure 2 is given by the sum of ${\mathrm{MDL}}^{\mathrm{online}}\left(A|C\right)$ and ${\mathrm{MDL}}^{\mathrm{online}}\left(B,A|C\right)$.

### 4.4. Detecting Adversarial Attacks

When an adversarial attack is conducted, it modifies a subset of tokens in an imperceptible way such that the performance of the language models (e.g., hate speech detection models) significantly degrades. By viewing adversarial attacks as the causal mechanisms between the original text and the altered text, we can quantify the algorithmic information of the adversarial attack using the method presented in the last section. In this context, *B* represents the tokens modified by the attack, *A* represents the original tokens of *B* before the attack, and *C* represents the unmodified tokens. By computing the MDL following the causal and anticausal directions shown in the causal graph in Figure 2, we expect to reveal the adversarial attack from the difference in the MDL values derived from both causal and anticausal directions.

However, there are two obstacles in directly applying the above method. First, the modified and original tokens share the same positions in the text, which is different from the previous case where A,B are disjoint subsets. Second, in practice the original text data are absent and only the altered text data are available, so the question is given a specified set of tokens: how to determine whether it has been modified or partially modified by the attack. To address these two issues, we propose to first generate a set of tokens as a proxy of the original tokens and compute the MDL between the generated tokens and the specified tokens. Specifically, given a specified set of tokens *B* and the context *C*, we generate the proxy tokens (denoted as $\hat{A}$) based on *C* using LLMs (we use the pre-trained BERT in our experiments). We then similarly mask the tokens in $\hat{A}$ with [*MASK*] tokens to obtain the input sequence $C^{in}$ for computing ${\mathrm{MDL}}^{\mathrm{online}}\left(\hat{A}|C\right)$ as well as the fine-tuned model $p_{\theta }^{f2}$. However, when computing $\mathrm{MDL}(B|\hat{A},C)$, if we fine-tune the model on *B*, the model will lose the information of $\hat{A}$ as $\hat{A},B$ share the same positions. Thus, in this case we directly evaluate the code-length of $p_{\theta }^{f2}$ without further fine-tuning it, resting on the belief that the fine-tuned model $p_{\theta }^{f2}$ already encompasses sufficient information to encode *B*. We expect the MDL values to reflect the algorithmic information of the attack as well in this case. In addition, we expect the MDL values to be proportional to the fraction of the modified tokens, serving as an indicator of detecting whether the specific set of tokens has been modified by the attack.

Based on the above proposed method for computing the MDL, we identify two key scenarios where the adversarial attacks can be effectively detected:

**Scenario 1.** In instances where a set of tokens can be verified to remain unaltered during the attack, we can calculate the MDL in a causal context for this unaltered token set to serve as a benchmark. Comparing this benchmark MDL against the MDL computed for other token sets enables the identification of potentially altered tokens. Here, the premise is that the benchmark set of unaltered tokens will exhibit a lower causal MDL compared to sets that include modified tokens.

**Scenario 2.** When a reference set of unchanged tokens is unavailable, an alternative approach involves randomly selecting subsets of tokens from the dataset to act as a comparative baseline. In this method, token sets demonstrating a lower causal MDL are inferred to be less affected by adversarial modifications, serving as an indirect means to gauge the extent of alteration within the data.

We outline the procedure for identifying potential adversarial modifications for both scenarios in Algorithm 1. Lines 3–13 in the algorithm aim to establish a benchmark MDL score. For Scenario 1, where the input comprises a specific set of unalteredTokens, the MDL score for this set is directly computed and utilized as the benchmark. On the other hand, for Scenario 2, where such a predefined set is absent, the algorithm generates random subsets of tokens from the dataset, calculates the MDL score for each subset, and subsequently identifies the subset with the lowest MDL score to serve as the benchmark. Following this, the algorithm computes the MDL score for the targetTokens on line 14. Lines 15–17 then evaluate whether these targetTokens have been subject to adversarial alterations, employing a predefined threshold δ to facilitate this determination.
**Algorithm 1** Detect Adversarial Tokens1:**Input:** textData, targetTokens, unalteredTokens (optional)2:**Output:** Identify if targetTokens contains adversarial modifications3:**if** unalteredTokens is available **then**4:    benchmarkMDL← CalculateMDL(*unalteredTokens*)5:**else**6:    randomSubsets← GenerateRandomSubsets(*textData*)7:    MDLScores←emptylist8:    **for** each subset in randomSubsets **do**9:        mdl← CalculateMDL(*subset*)10:        Append (subset,mdl) to MDLScores11:    **end for**12:    benchmarkMDL←min(MDLScores,key=mdl).mdl13:**end if**14:mdlScore← CalculateMDL(*targetTokens*)15:**if** IsAdversariallyModified(benchmarkMDL, mdlScore, δ) **then**16:    **Ouput** TRUE17:**else**18:    **Ouput** FALSE19:**end if**

We present in the next section the experiment results that support our method.

## 5. Experiments

### 5.1. Datasets

To conduct experiments for empirical evaluations, we adopt the Amazon Review Polarity dataset [43] as our base dataset. This dataset is pre-labeled and is widely used for sentiment analysis that aims to categorize reviews into negative and positive reviews. In the raw dataset, there are 5 review sources where scores 1 and 2 are considered positive, scores 4 and 5 are considered negative, and score 3 is excluded. As a pre-processing step, we convert the labels into binary labels with negative and positive sentiments. Then, we randomly sample 20,000 instances from this dataset, pre-process it to remove any unnecessary tokens and characters, and obtain the base dataset denoted as $D^{org}$. We will later adapt this dataset to cater to the various requirements of our experimental scenarios.

### 5.2. Evaluating MDL between Tokens in Original Text

First, we evaluate whether any clear causal relationships exist between different token subsets that might affect our subsequent analysis. For this purpose, we randomly select two disjoint subsets of tokens as *A* and *B* and compute the MDL following the causal graph in Figure 2 for both causal and anticausal directions. Specifically, we randomly select five different subsets as *A*, and for each *A* we further randomly select three different subsets as *B*, resulting in 15 different pairs of A,B. The MDL values computed are shown in Table 1, where causal MDL is computed as MDL(A|C)+MDL(B|A,C) and anticausal MDL is computed as MDL(B|C)+MDL(A|B,C). As can be seen, the MDL values from both directions are nearly identical. The average confidence score S=0.0013 suggests that there is no clear causal relation between these two sets of tokens in the original text.

### 5.3. Evaluating MDL between Tokens in Original and Altered Text

Then, we perform the adversarial attack and evaluate the MDL between tokens modified by the attack and their original counterparts. We employ two adversarial attacks, BERT-ATTACK and TextFooler, to select a subset of tokens in each review to be altered to potentially flip the model’s predicted label for that review. Specifically, for each review $X_{i}\in D^{org}$, we denote $m_{i}$ as a list of vulnerable tokens selected by the adversarial attack such that $m_{i}=\left\{t_{1},t_{2},\ldots ,t_{n}\right\}$ and $M=\{m_{1},m_{2},\ldots ,m_{i}\}$ is a list of all the vulnerable tokens in $D^{org}$. Then, instead of modifying all tokens in *M*, we randomly choose 30% of the tokens in $m_{i}$ for each review and modify them according to the adversarial attack while the remaining 70% tokens were kept in their original form. In this way we can repeat the experiment mutiple times with different modified tokens. In the experiments, we repeat the experiment five times. We denote the modified dataset as $D^{mod}$. The modified tokens and their originals are denoted as *B* and *A* to maintain consistent notation.

We then compute the MDL similarly according to the causal graph in Figure 2. Since *A* and *B* share the same positions, we adopt the adapted method presented in Section 4.4. The results are shown in Table 2 and Table 3, from which we see that the causal MDL is consistently smaller than the anticausal MDL. As Table 1 shows no clear causal relations between tokens in the original text, the difference in the MDL values is likely to stem from the difference in the difficulty of inference between the causal and anticausal directions. For example, given a vulnerable token that is selected by the adversarial attack, we can infer the modified token by encoding the mechanism of the adversarial attack model. Vice versa, it is more difficult to infer the original token based on the modified token, which requires reversing the mechanism of the adversarial attack. We may also interpret this situation based on the principle of independent mechanisms; i.e., given *t* as a vulnerable token selected by the adversarial attack, the modified counterpart of *t* is independent of the distribution of *t* in the dataset. Thus, the results in Table 2 and Table 3 can be considered as evidence to support the hypothesis that the algorithmic information of the adversarial attack model can be captured by computing the MDL values.

### 5.4. Evaluating MDL for Detecting Adversarial Attacks

Finally, we evaluate our approach for detecting adversarial attacks given $D^{mod}$ only, where we employ the base BERT to generate the tokens for MDL computation. Specifically, given *M*, the vulnerable tokens, we similarly choose 30% of the tokens to modify and denote them as *B*. Then, we mask the tokens in *B* by [*MASK*] and use the base BERT to generate tokens in the corresponding positions, denoted as $\hat{A}$, for computing $\mathrm{MDL}(\hat{A},B|C)$. To repeat the experiment, we randomly select five different *B*s and for each *B* we generate three different $\hat{A}$s. The results of all 15 runs are shown in Table 4 for BERT-ATTACK and Table 5 for TextFooler. As can be seen, the causal and anticausal MDL values are similar to the results in Table 2, where the original tokens are used in the computation. This result implies that the generated tokens could be used as proxies in estimating the MDL between tokens modified by the attack and their original counterparts, which may reflect the algorithmic information of the attacking model.

To further assess our idea, we construct multiple *B*s that involve different percentages of the modified tokens and evaluate how the MDL values vary with the change of the fraction of modified tokens. We first randomly select 30% tokens from *M* as *B*. Then, we construct $B_{p}$ by randomly selecting p%·|B| tokens from *B* that are truly modified by the attacking model and (1−p%)·|B| from *M* that are not modified. Using this method, we create $B_{0}$, $B_{25}$, $B_{50}$, $B_{75}$, and $B_{100}$ where $B_{100}$ is equivalent to *B*. Then, we generate ${\hat{A}}_{p}$ similarly for each $B_{p}$ and compute $\mathrm{MDL}({\hat{A}}_{p},B_{p}|C)$. We repeat the experiment 15 times for each *p* and present the results in Figure 3. We can see that the causal MDL rises with an increasing percentage of modified tokens for both attacking models. This is as expected according to our hypothesis that the MDL reflects the algorithmic information of the attacking model, since a greater number of modified tokens would require a longer code-length to compress the attack mechanism.

However, we notice that the anticausal MDL does not follow a consistent trend for different attacking models. We presume that this is due to the anticausal MDL being primarily influenced by the difficulty of predicting tokens generated by the base BERT. To verify this idea, we generate tokens based on the original text and compute the MDL for the generated tokens $\hat{A}$ and the original tokens *B*. The results are shown in Table 6. We see that without any modification, the causal MDL remains prominently smaller than the anticausal MDL, which verifies our conjecture to a certain extent.

## 6. Conclusions

In this paper, we investigated the use of MDL to analyze the causal influence caused by adversarial attacks. We developed techniques for computing the MDL for inferring causal directions between tokens by using MLM. Based on that, we developed techniques for judging whether a specified set of tokens has been altered by the attack, even in the absence of the original text data. In the experiments, we showed a clear difference in the causal and anticausal MDL values, implying that algorithmic information of the attacking model can be captured by computing the MDL values. In addition, we showed that by generating proxy tokens, the causal MDL values are proportional to the fraction of the modified tokens in a specified subset, which could serve as an indicator of detecting whether the tokens have undergone modification. Our work made a preliminary effort to identify adversarial attacks and has the potential to enhance the applicability of existing robust NLP models.

## Figures and Tables

**Figure 1 entropy-26-00354-f001:**
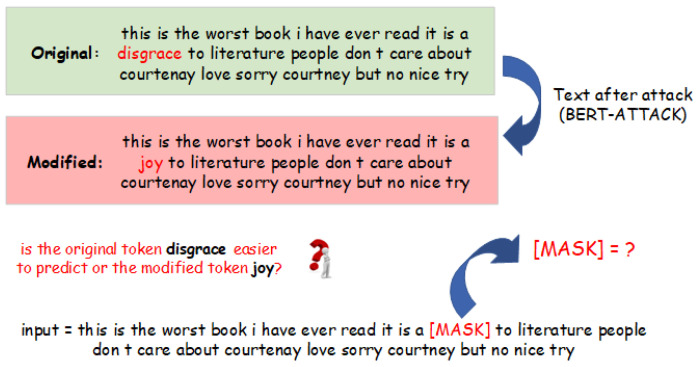
Procedure of MDL computation with masked language modeling (MLM): An initial review is sourced from the original dataset and subjected to the BERT-ATTACK module, resulting in the generation of an adversarial review. MLM is employed to evaluate the ‘amount of effort’ of prediction for two categories of tokens: the original token and the modified token. Subsequently, the minimum description length (MDL) is computed based on these evaluations.

**Figure 2 entropy-26-00354-f002:**
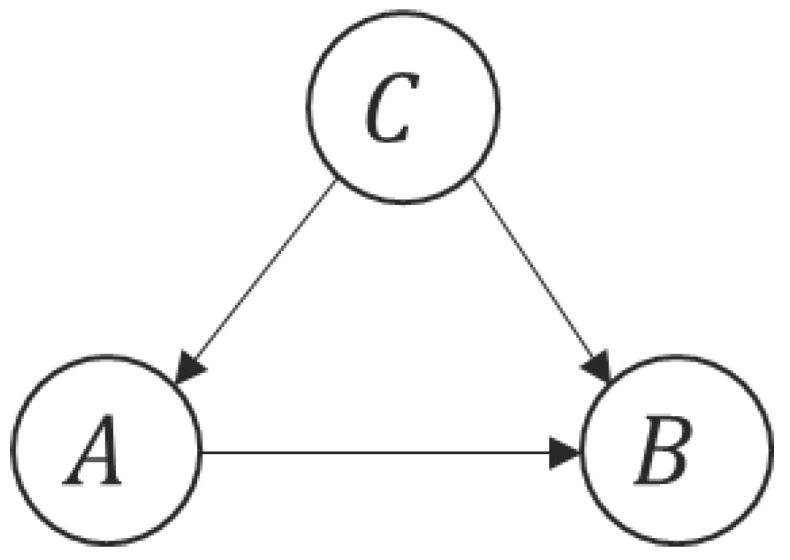
Causal graph for showing different components of a text in a causal relationship.

**Figure 3 entropy-26-00354-f003:**
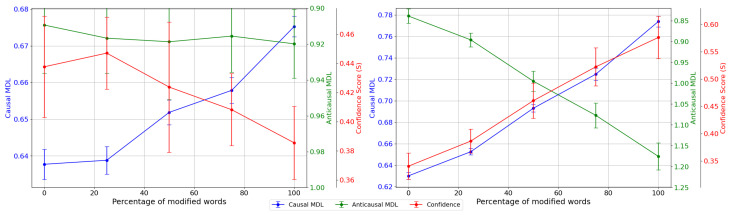
MDL for causal direction, anticausal direction, and confidence score (S) of MDL with standard deviation for different percentages of modified words of BERTAttack (**left**) and TextFooler (**right**).

**Table 1 entropy-26-00354-t001:** MDL between tokens in the original text.

ID	MDL(A|C)	MDL(B|A,C)	MDL(B|C)	MDL(A|B,C)	Causal	Anticausal	$\Delta_{A\to B}$	$\Delta_{B\to A}$	$S=|\Delta_{A\to B}-\Delta_{B\to A}|$
1	0.3653	0.3695	0.3664	0.3686	0.7348	0.7350	1.0042	1.0045	0.0003
2	0.3658	0.3689	0.3662	0.3691	0.7347	0.7353	1.0037	1.0045	0.0008
3	0.3651	0.3695	0.3661	0.3684	0.7346	0.7345	1.0046	1.0045	0.0001
4	0.3648	0.3677	0.3654	0.3685	0.7325	0.7339	1.0031	1.0051	0.0019
5	0.3659	0.3682	0.3662	0.3695	0.7341	0.7357	1.0027	1.0049	0.0022
6	0.3652	0.3694	0.3660	0.3689	0.7346	0.7349	1.0046	1.0051	0.0004
7	0.3658	0.3689	0.3653	0.3692	0.7347	0.7345	1.0049	1.0047	0.0003
8	0.3650	0.3683	0.3652	0.3684	0.7333	0.7336	1.0042	1.0047	0.0004
9	0.3657	0.3695	0.3661	0.3680	0.7352	0.7341	1.0046	1.0031	0.0015
10	0.3651	0.3692	0.3656	0.3692	0.7343	0.7348	1.0049	1.0056	0.0007
11	0.3646	0.3679	0.3664	0.3698	0.7325	0.7362	1.0021	1.0071	0.0051
12	0.3658	0.3684	0.3652	0.3695	0.7342	0.7347	1.0044	1.0051	0.0007
13	0.3657	0.3679	0.3651	0.3694	0.7336	0.7345	1.0038	1.0051	0.0012
14	0.3660	0.3699	0.3658	0.3698	0.7359	0.7356	1.0056	1.0052	0.0004
15	0.3661	0.3689	0.3646	0.3678	0.7350	0.7324	1.0059	1.0023	0.0036
**Mean**	0.36546	0.36881	0.36571	0.36894	0.73427	0.73465	1.00422	1.00477	0.00131
**±**	±	±	±	±	±	±	±	±	±
**SD**	0.00045	0.00067	0.00053	0.00061	0.00092	0.00090	0.00099	0.00103	0.00136

**Table 2 entropy-26-00354-t002:** MDL between the tokens in the original and altered text (BERT-ATTACK).

ID	MDL(A|C)	MDL(B|A,C)	MDL(B|C)	MDL(A|B,C)	Causal	Anticausal	$\Delta_{A\to B}$	$\Delta_{B\to A}$	$S=|\Delta_{A\to B}-\Delta_{B\to A}|$
1	0.3606	0.3617	0.3362	0.5674	0.7223	0.9036	1.0400	1.3000	0.2600
2	0.3659	0.3606	0.3363	0.5823	0.7265	0.9186	1.0300	1.3100	0.2800
3	0.3669	0.3623	0.3365	0.5826	0.7292	0.9191	1.0400	1.3100	0.2700
4	0.3662	0.3618	0.3364	0.5685	0.7280	0.9049	1.0400	1.2900	0.2500
5	0.3720	0.3603	0.3365	0.5754	0.7323	0.9119	1.0300	1.2900	0.2600
**Mean**	0.36632	0.36134	0.33638	0.57524	0.72766	0.91162	1.03600	1.30000	0.26400
**±**	±	±	±	±	±	±	±	±	±
**SD**	0.00362	0.00076	0.00012	0.00650	0.00329	0.00655	0.00490	0.00894	0.01020

**Table 3 entropy-26-00354-t003:** MDL between the tokens in the original and altered text (TextFooler).

ID	MDL(A|C)	MDL(B|A,C)	MDL(B|C)	MDL(A|B,C)	Causal	Anticausal	$\Delta_{A\to B}$	$\Delta_{B\to A}$	$S=|\Delta_{A\to B}-\Delta_{B\to A}|$
1	0.4683	0.3614	0.324	1.6700	0.8297	1.9940	1.0500	2.5200	1.4700
2	0.4705	0.3730	0.3244	2.4532	0.8435	2.7776	1.0600	3.4900	2.4300
3	0.4686	0.3702	0.3235	1.1177	0.8388	1.4412	1.0600	1.8200	0.7600
4	0.4704	0.3833	0.3239	2.2307	0.8537	2.5546	1.0700	3.2200	2.1500
5	0.4741	0.3657	0.3242	2.2099	0.8398	2.5341	1.0500	3.1700	2.1200
**Mean**	0.47038	0.37072	0.32400	1.93630	0.84110	2.26030	1.05800	2.84400	1.78600
**±**	±	±	±	±	±	±	±	±	±
**SD**	0.00207	0.00743	0.00030	0.48368	0.00776	0.48395	0.00748	0.60308	0.60188

**Table 4 entropy-26-00354-t004:** MDL between the modified tokens and their generated counterparts (BERT-ATTACK).

ID	MDL(A|C)	MDL(B|A,C)	MDL(B|C)	MDL(A|B,C)	Causal	Anticausal	$\Delta_{A\to B}$	$\Delta_{B\to A}$	$S=|\Delta_{A\to B}-\Delta_{B\to A}|$
1	0.3361	0.3400	0.3000	0.6487	0.6761	0.9487	1.0629	1.4914	0.4285
2	0.3330	0.3420	0.3000	0.6157	0.6750	0.9157	1.0664	1.4466	0.3803
3	0.3354	0.3421	0.3001	0.6243	0.6775	0.9244	1.0661	1.4546	0.3885
4	0.3334	0.3399	0.2996	0.6119	0.6733	0.9115	1.0637	1.4400	0.3763
5	0.3344	0.3397	0.2996	0.6070	0.6741	0.9066	1.0632	1.4300	0.3667
6	0.3369	0.3394	0.2994	0.6232	0.6763	0.9226	1.0629	1.4499	0.3871
7	0.3338	0.3404	0.2998	0.6275	0.6742	0.9273	1.0641	1.4635	0.3995
8	0.3356	0.3421	0.2996	0.6265	0.6777	0.9261	1.0669	1.4580	0.3911
9	0.3340	0.3405	0.2996	0.6175	0.6745	0.9171	1.0646	1.4474	0.3829
10	0.3275	0.3419	0.3000	0.5689	0.6694	0.8689	1.0668	1.3847	0.3179
11	0.3374	0.3409	0.3002	0.6350	0.6783	0.9352	1.0638	1.4668	0.4029
12	0.3367	0.3412	0.3000	0.6274	0.6779	0.9274	1.0647	1.4566	0.3919
13	0.3299	0.3403	0.3004	0.5998	0.6702	0.9002	1.0633	1.4282	0.3649
14	0.3343	0.3405	0.3003	0.6156	0.6748	0.9159	1.0633	1.4433	0.3799
15	0.3391	0.3407	0.3003	0.6508	0.6798	0.9511	1.0632	1.4875	0.4243
**Mean**	0.33450	0.34077	0.29993	0.61999	0.67527	0.91991	1.06439	1.44990	0.38551
**±**	±	±	±	±	±	±	±	±	±
**SD**	0.00282	0.00087	0.00030	0.01911	0.00279	0.01913	0.00141	0.02452	0.02508

**Table 5 entropy-26-00354-t005:** MDL between the modified tokens and their generated counterparts (TextFooler).

ID	MDL(A|C)	MDL(B|A,C)	MDL(B|C)	MDL(A|B,C)	Causal	Anticausal	$\Delta_{A\to B}$	$\Delta_{B\to A}$	$S=|\Delta_{A\to B}-\Delta_{B\to A}|$
1	0.3812	0.3966	0.324	0.9288	0.7778	1.2528	1.1029	1.7765	0.6736
2	0.3763	0.4012	0.3241	0.8416	0.7775	1.1657	1.1101	1.6643	0.5543
3	0.3719	0.3998	0.324	0.8603	0.7717	1.1843	1.1089	1.7018	0.5929
4	0.3742	0.4021	0.3244	0.8864	0.7763	1.2108	1.1112	1.7332	0.6220
5	0.37	0.3999	0.3244	0.8381	0.7699	1.1625	1.1087	1.6741	0.5654
6	0.3694	0.4009	0.3243	0.8432	0.7703	1.1675	1.1104	1.6830	0.5726
7	0.3725	0.4025	0.3235	0.8231	0.7750	1.1466	1.1135	1.6474	0.5339
8	0.379	0.3988	0.3235	0.888	0.7778	1.2115	1.1072	1.7246	0.6174
9	0.3705	0.4005	0.3235	0.8305	0.7710	1.1540	1.1110	1.6628	0.5519
10	0.3652	0.4019	0.3239	0.8162	0.7671	1.1401	1.1132	1.6545	0.5413
11	0.3735	0.4008	0.3241	0.866	0.7743	1.1901	1.1099	1.7060	0.5960
12	0.365	0.3984	0.3242	0.804	0.7634	1.1282	1.1077	1.6370	0.5293
13	0.3825	0.4005	0.3242	0.8632	0.7830	1.1874	1.1080	1.6802	0.5722
14	0.3782	0.4007	0.3239	0.8658	0.7789	1.1897	1.1094	1.6945	0.5851
15	0.3707	0.4013	0.3242	0.8165	0.7720	1.1407	1.1110	1.6415	0.5306
**Mean**	0.37334	0.40039	0.32401	0.85145	0.77373	1.17546	1.10954	1.68543	0.57589
**±**	±	±	±	±	±	±	±	±	±
**SD**	0.00511	0.00149	0.00030	0.03212	0.00490	0.03212	0.00249	0.03713	0.03875

**Table 6 entropy-26-00354-t006:** MDL between the original tokens and their generated counterparts.

ID	MDL(A|C)	MDL(B|A,C)	MDL(B|C)	MDL(A|B,C)	Causal	Anticausal	$\Delta_{A\to B}$	$\Delta_{B\to A}$	$S=|\Delta_{A\to B}-\Delta_{B\to A}|$
1	0.3361	0.3405	0.3192	0.454	0.6766	0.7732	1.0300	1.1800	0.1500
2	0.3334	0.3444	0.3128	0.4141	0.6778	0.7269	1.0500	1.1200	0.0700
3	0.3338	0.3605	0.3171	0.4206	0.6943	0.7377	1.0700	1.1300	0.0600
4	0.3275	0.3519	0.3131	0.3846	0.6794	0.6977	1.0600	1.0900	0.0300
5	0.3299	0.3514	0.315	0.4004	0.6813	0.7154	1.0600	1.1100	0.0500
**Mean**	0.33214	0.34974	0.31544	0.41474	0.68188	0.73018	1.054	1.126	0.072
**±**	±	±	±	±	±	±	±	±	±
**SD**	0.00305	0.00689	0.00243	0.02320	0.00641	0.02526	0.01356	0.03007	0.04118

## Data Availability

All the code and data used in this work are available at https://github.com/zthsk/MDL-MLM (access on 1 January 2024).

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
