# Peer review of "Implications of Minimum Description Length for Adversarial Attack in Natural Language Processing"

_entropy, 2024, doi:10.3390/e26050354_

Round 1
Reviewer 1 Report
Comments and Suggestions for Authors
I find the topic of the paper highly interesting, and the idea of using MDL on this problem is compelling. However, I don’t think the paper is well written. I think it relies too much on referencing other papers for things that could easily have been described in the paper. The paper should be written more concisely and precisely while also includin more details.
First, it is not clear from the paper exactly why MDL is appropriate for adversarial attacks. I guess it because of ICM and the relationship in Fig. 2. But it is never made quite clear except perhaps through the equation in line 279.
Following this, there are many things that is simply not comprehensible from the text. There is equation (1) without any explanation of the symbols in the equation. There is equation 2: what does (X,Y) mean? Codelength of encoding (X,Y)? With an optimum code? With a universal code? The equation is clearly correct in terms of entropy, but not for codelength of individual sequences. Similarly, the equation in line 279 has equality in terms of entropy. Then why is there inequality? Since the notation ( ) has not been defined this cannot be verified. It is clear this equation comes from [16] (although no reference is given here), and in [16] it is argued in terms of Kolmogorov complexity. But then it only holds except for a constant term, as is usual in Kolmogorov theory. The way this inequality is argued in [16] is by replacing Kolmogorov complexity with MDL, or universal codelength. That is a fair argument, although also there it is not clear what the inequality means: true except for a constant? Still, in this paper the inequality is not even argued, and because the notation ( ) is not defined, it is not even clear what could be the meaning.
The meaning of equation (3) is not clear either; at least here there is an explanation as opposed to equation (1). But the explanation is very imprecise. Only be reading [42] does it become clear that this is a sequential block coder. I would say the explanation before (3) does nothing for people unfamiliar with MDL and only confuses people familiar with MDL.
Finally, in Section 4.4 there is no precise description of how adversarial attacks are detected. Are they detected by some quantity>threshold? Or be some measure of adversarity? Some equations to make this precise would help.
In line 150 the equation contains y_{i,j} but the text only explains y_i.
In conclusion, I think the method in the paper might be interesting, but the writing must be improved. Right now the writing is verbose rather than concise, and lacks precision. More concretely, I would recommend more equations and less words.
Reviewer 2 Report
Comments and Suggestions for Authors
NLP adversarial attack is an important part of artificial intelligence adversarial attack, which is different from traditional CV or audio adversarial attack, because its value is composed of discrete tokens one by one, which has a greater challenge. This work studies the robust NLP problem from a complementary but different angle, innovatively regards the behavior of the attack model as a complex causal mechanism, and uses the minimum description length (MDL) framework to quantify its algorithmic information, achieving good experimental results.
The article is well-written and organized. In order to highlight the superiority of the algorithm proposed in this paper, it is suggested to increase the comparison and analysis with the classical adversarial attack model in NLP domain.
Round 2
Reviewer 1 Report
Comments and Suggestions for Authors
"Finally, in Section 4.4 there is no precise description of how adversarial attacks are detected. Are they detected by some quantity>threshold? Or be some measure of adversarity? Some equations to make this precise would help."
The authors did add additional text, but I still feel the paper lacks precision in Section 4.4. Either some equation to make precise how attacks are detected, or some pseudo code, would be helpful. Here I feel it’s critical to be more precise.
Author Response
Thank you for your feedback on Section 4.4 of our manuscript. We have taken your comments into account and improved the clarity in describing how adversarial attacks are detected. Specifically, we've added a detailed pseudocode (Algorithm 1) and an accompanying explanation to the section. This update aims to clearly outline the process of identifying attacks for different scenarios.
Round 3
Reviewer 1 Report
Comments and Suggestions for Authors
I am OK with the paper now.